# Quantifying the Widths of Fault Damage Zones Based on the Fault Likelihood: A Case Study of Faults in the Fuji Syncline of the Luzhou Block, Sichuan Basin, China

**Lu Zeng [1], Jinxi Li [2,*], Shihu Wu [3], Kailin Tong [4] and Zhiwu Li [5]**

[1]  Key Laboratory of Mountain Hazards and Earth Surface Processes, Institute of Mountain Hazards and Environment, Chinese Academy of Sciences, Chengdu 610299, China; zenglu@imde.ac.cn
[2]  Key Laboratory of Earth Exploration and Information Technology of Ministry of Education, Chengdu University of Technology, Chengdu 610059, China
[3]  Exploration and Development Research Institution of Southwest Oilfield Branch of China Petroleum Company, Chengdu 610041, China
[4]  Sichuan Shale Gas Exploration and Development Company Limited, Chengdu 610051, China
[5]  State Key Laboratory of Oil and Gas Reservoir Geology and Exploitation, Chengdu University of Technology, Chengdu 610059, China
*   Correspondence: lijinxi07@cdut.edu.cn; Tel.: +86-13666299365

**Abstract:** Faults are critical to the preservation or destruction of shale gas concentration. The Lower Silurian Longmaxi Formation in the southern Sichuan Basin hosts relatively developed faults, which pose a huge challenge to the exploration and exploitation of shale gas. An urgent need to quickly determine the widths of fault damage zones (FDZs) arises in locating horizontal shale gas wells. In this study, FDZs were estimated using the fault likelihood. The results are as follows: (1) It is rational to constrain the FDZ width using a fault likelihood greater than 0.2. The six major NEE-trending faults in the Fuji syncline of the Luzhou block have complex structures and varying FDZ widths from about 240–1220 m. (2) The degree of influence of FDZs is negatively correlated with their distance from the faults. In other words, a greater distance from a fault is associated with a weaker influence and a smaller fault likelihood. (3) Based on the ratio of the fault throw to the FDZ width, we propose that the width of seismic-scale fault damages can be directly constrained using a ratio value of 3.5. This method is fast and accurate and can provide support for the evaluation of the shale gas preservation conditions and well placement in the Longmaxi Formation of the southern Sichuan Basin.

**Keywords:** fault damage zone; fault throw; fault likelihood; shale gas; Sichuan Basin

## 1. Introduction

Brittle faults are one of the most important structures that are widely developed in the shallow crust. Faulting causes a local concentration of stress and strain of rocks, provides favorable migration and precipitation spaces for underground fluid motions, and drives the migration of fluids such as hydrocarbons, ore-forming fluids, and groundwater, thus controlling the accumulation and distribution of fluids [1–3]. Therefore, faults have always been an important basic research subject in geology. Previous researchers have put great efforts into the characterization of the geometric features of faults, including observation and description of the geometric elements of faults, as well as statistical analysis of the dip angle, length, and throw of faults, the FDZ width, the fabric of faulted rocks, and the development of fault damages [4–8]. Moreover, they characterized quantitative relationships among some fault elements, thereby laying a solid foundation for the study of the formation process and origin of faults. However, the results of previous studies were mostly based on surface fault zones, and there is a lack of studies on fine-scale characterization of subsurface

faults and the quantitative relationships among various fault elements. Therefore, previous studies fail to meet the actual demand for hydrocarbon exploration.

Faults act as important pathways for the migration, accumulation, and expulsion of hydrocarbons and are a critical factor in the formation of hydrocarbon reservoirs, especially conventional hydrocarbon reservoirs [9–11]. Depending on the influencing scope of faulting, faults are not always catastrophically destructive to continuously and planarly distributed shale gas. Fault damages are the most direct product of faulting, and their geometric morphologies and developmental degrees can reflect the properties, scales, tectonic parts, and rock mechanical properties of faults [8,12]. The analysis of surface structures allows for effective constraining of the scope of fault zones and the developmental degrees of fractures on both sides of the fault zones and even direct quantification of some important geometric parameters of fault damages [4,6,7]. However, for concealed faults in basins, the occurrence of fault damages and the influencing scope of faulting are, at present, primarily revealed using geophysical data or geophysical data combined with drilling data. In addition, no effective quantitative parameters or indices are available to evaluate the preservation conditions of shale gas or guide the optimization of well deployment [13,14].

The southern Sichuan Basin is one of the areas of China with the most abundant shale gas resources since the Wufeng–Longmaxi formations in this region present relatively thick and stably distributed high-quality shales [15–19]. However, the southern Sichuan Basin holds complex structures; fractures on the hanging and foot walls of faults in this region have different developmental degrees and the faults have varying FDZ widths. Moreover, the production capacity of shale gas in this region differs significantly as the distance from faults varies. These factors need to be urgently solved to facilitate the exploration and exploitation of shale gas in the southern Sichuan Basin. Based on the fault likelihood of seismic data, we characterized the structural morphology of the major NNE-trending FDZs in the Fuji syncline of the Luzhou block, southwestern Sichuan Basin, and conducted a quantitative analysis of the FDZ widths. Accordingly, this study provides a new method and tool for analyzing the internal structures of shale FDZs and thus is of great significance in research on regional shale gas accumulation systems.

## 2. Basic Structures of FDZs

The intensity of deformations caused by the action of faulting on rocks gradually decreases from the cores toward both sides of the fault. Based on the fabric characteristics of faulted rocks and the developmental degrees of fractures, a complete fault damage zone (FDZ) can be divided into a fault core and damage zone on both sides [4,20,21]. The fault core, which exhibits the most concentrated strain in a fault, consists primarily of fault gouge, cataclasites, and breccias, with pseudotachylites possibly occurring locally in the fault core. Moreover, cleavages are highly developed in the fault core. In the damage zone, the strain and the developmental degrees of tectonic foliations significantly decrease from the fault core to both sides. The rocks in FDZs largely maintain their initial fabrics but show significant cleavages, with the fracture density gradually decreasing toward both sides (Figure 1). As the distance from the fault core increases, the surrounding rocks tend to be intact, structures associated with faults gradually weaken, and fractures disappear gradually. The joint planes and fractures of rocks are mainly controlled by the regional stress field [22–24].

The FDZ width of a fault, or the influencing scope of faulting, is primarily characterized based on the developmental degrees of fractures at a straight-line distance from the fault [9,25]. Commonly used statistical methods for fault damages include: (1) 1D linear scanning: fault damages are quantified using the number of fractures per unit length [26,27]; (2) statistical analysis of 2D areal density: fault damages are quantified using the length of fractures per unit area of a circle, square, or triangle [26,28]; (3) 2D circular scanning: fault damages are quantified using the number of fracture endpoints within a unit circle or the number of intersection points between fractures and the edge of a unit circle [26,29–31].

These methods have their respective applicability, advantages, and disadvantages and should be employed according to the actual situation.

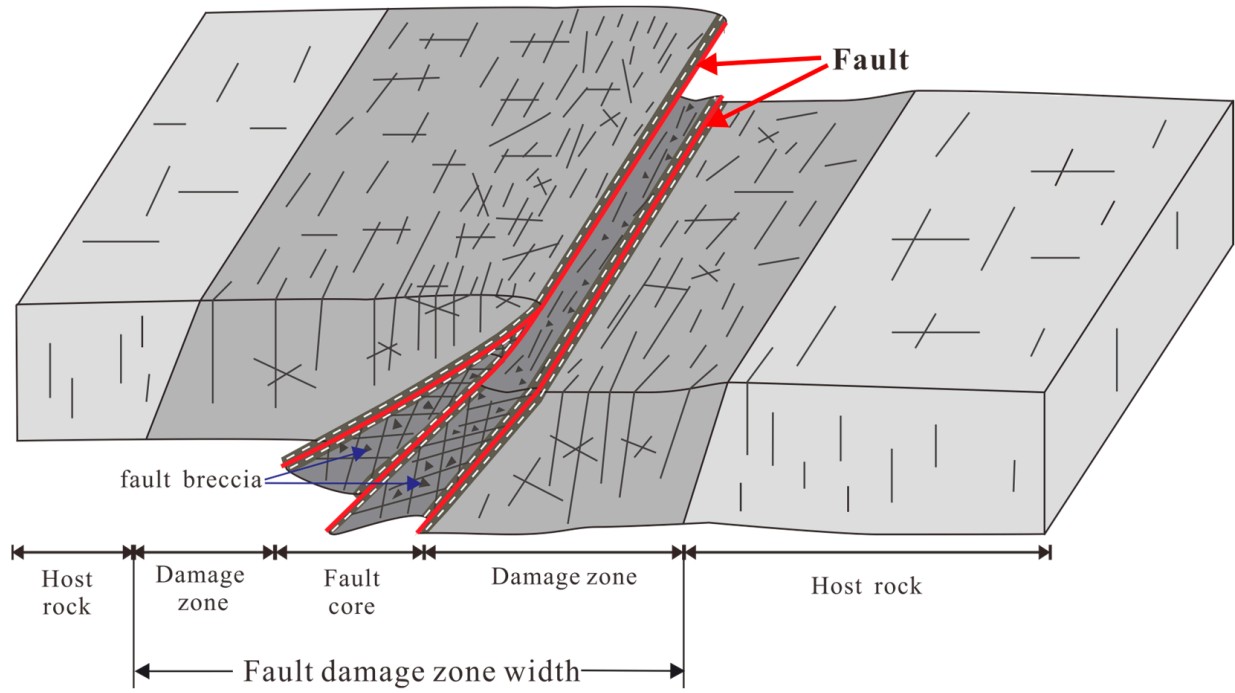

**Figure 1.** Fault zone architecture and fractures in each component.

Many researchers have produced detailed descriptions and characterizations of exposed faults through structural analysis, determining the geometric parameters of fault damages and their relationships [4,6,7,32]. Generally, fractures are densely developed in fault cores and FDZs, and their density decreases exponentially as the distance from fault cores increases until it equals the background value of fractures in the surrounding rocks on both sides [33]. For instance, Fossen et al. [6] investigated a geological section perpendicular to the fault strike. They analyzed the statistics of fractures in four nearly horizontal strata on hanging walls of faults and characterized the relationship between the fracture density and the distance from faults (Figure 2a,b). According to the relationship, the aeolian siltstones on the top and bottom had a much higher fracture density than the fluvial siltstones in the middle part, with the declining rate of the fracture density of the top and bottom being significantly lower than that of the middle part [6]. Specifically, the aeolian siltstones had a fracture density of less than three fractures per meter (i.e., the background value of the fracture density of the surrounding rocks in a single horizontal stratum) at a distance of 50 m and above from the fault cores, while the fluvial siltstones had a fracture density of less than three fractures per meter at a distance of 10 m and above from the fault cores (Figure 2a,b). This result may reflect that the aeolian siltstones had a much lower degree of cementation than the fluvial siltstones, indicating that the fracture density is closely related to the mechanical properties of rocks. In addition, Fossen [32] conducted statistical analysis of the fractures in nearly horizontal sandstones on the foot wall of a fault zone composed of two normal faults F2 and F3 in the Moab area, Utah, finding a similar exponential relationship between the fracture density and the distance from the fault core (Figure 2c,d). Specifically, the fracture density decreased from 120 fractures per meter to less than 50 fractures per meter with an increase in the distance from the core of F2, increased to about 120 fractures per meter near F3, and then decreased to less than 5 fractures per meter (i.e., the background value of the fracture density of the surrounding rocks in a single horizontal stratum) at a distance of 15 m from the core of F3 (Figure 2c,d).

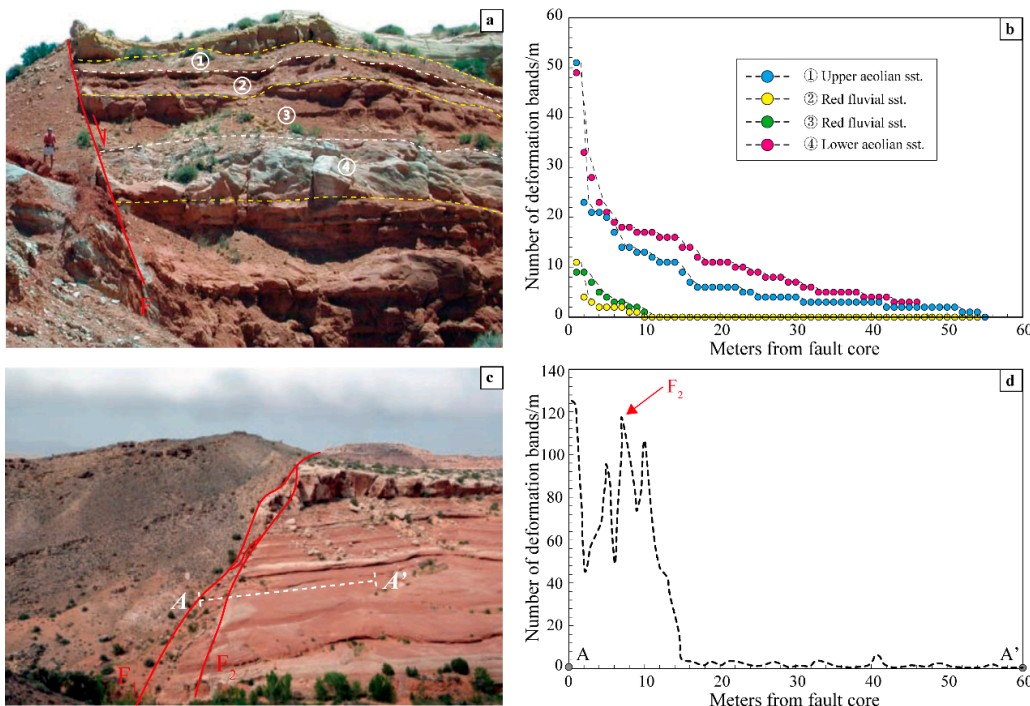

**Figure 2.** Relationship between horizontal distance from the fault core and fracture frequency. (**a**,**b**) Fracture frequency of horizontal strata in the foot wall and fracture density [6]. (**c**,**d**) Fracture frequency of horizontal strata in the hanging wall and fracture density [32].

In addition, previous researchers investigated FDZ widths of different types of faults using different methods (including surface and core surveys). They quantified the FDZ widths based on the developmental degrees of fractures. As shown with statistics, the Castle Cove Fault in the Otway Basin, Victoria, Australia, a reverse fault, had a density of surface fractures of up to a maximum of 510 fractures per 10 m at a distance of about 60 m from the fault core and this decreased to 120–140 fractures per 10 m (i.e., the background value of the fracture density of the surrounding rocks) at a distance of about 290 m from the fault core (Figure 3a) [7]. Therefore, the threshold for the FDZ width was 0.23–0.26 (i.e., the ratio of the background value to the maximum value; the dotted line in Figure 3a). According to the statistics of the surface fractures (Figure 3b), the well-exposed, nearly NW–SE-trending Moab Fault in Utah, USA, a normal fault, had a maximum fracture density of about 26 fractures per meter [4]. Since the background value of the fracture density at a distance of about 40 m from the fault core was 4–5 fractures per meter (the dotted line in Figure 3b), the threshold for the FDZ width was 0.15–0.19. As shown by the statistics of the fractures in cores drilled from a dextral strike-slip fault in Yangsan City, South Korea, the fracture density near the fault core was 30 fractures per 3 m and its background value at a distance of about 275 m from the main fault was 6–7 fractures per 3 m (the dotted line in Figure 3c). Therefore, the threshold for the FDZ width was 0.20–0.23 (Figure 3c). Liao et al. [34] carried out a quantitative analysis of the FDZ width of tight sandstones based on the variances of 3D seismic data. They selected the variance of 0.2 as the threshold based on the points subjected to changes in the slope of the cumulative variance. Accordingly, they determined that the FDZ width was 700–1200 m, which is roughly consistent with the result obtained based on the surface structures. The FDZ width in different strata is obviously different in the same fault (Figure 3d). As indicated by massive measured data of faults, for most faults, the ratios of the FDZ width to the distance from faults fall within the range between two lines corresponding to the ratios of 0.001 and 1 (Figure 3e).

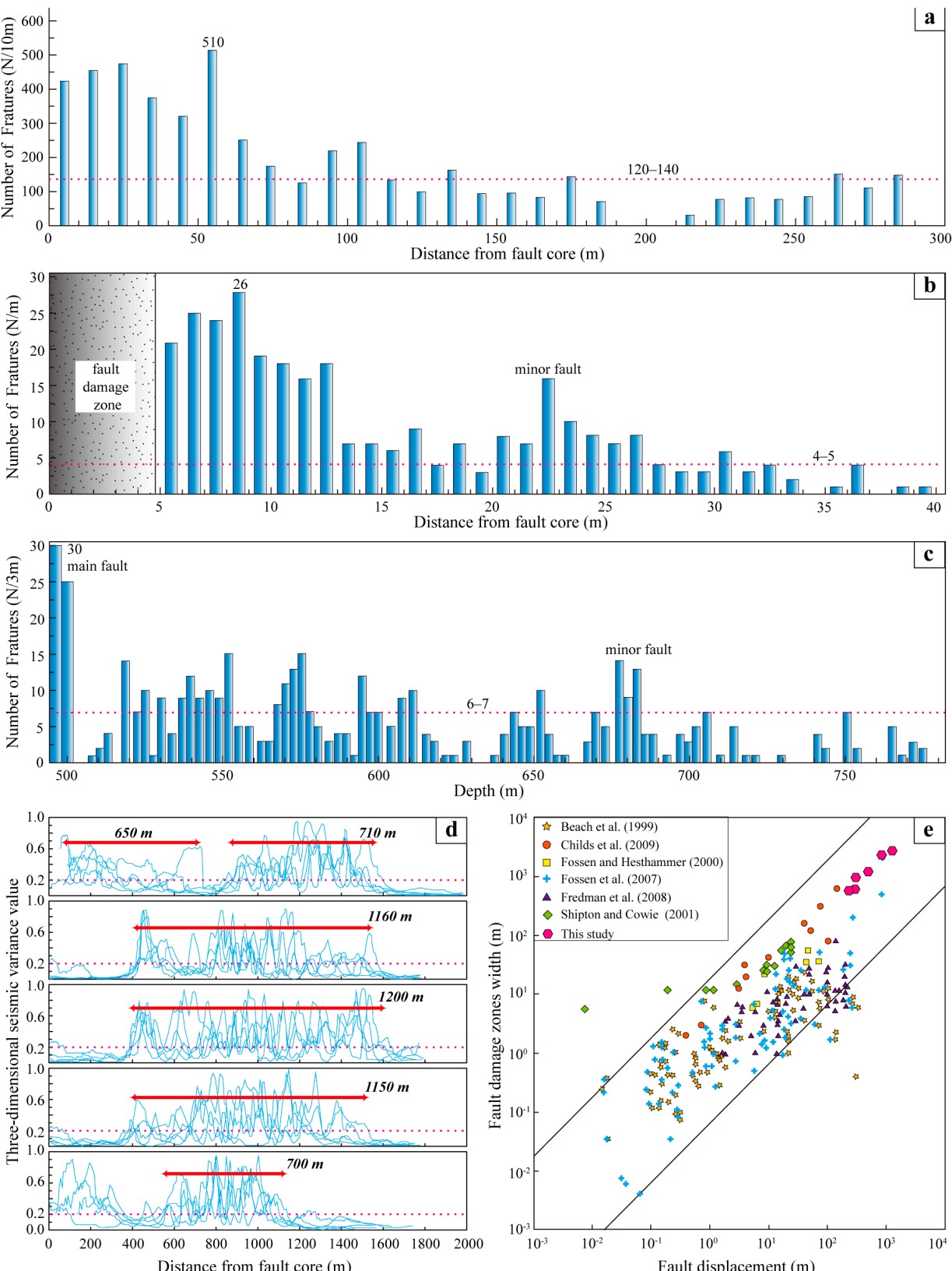

**Figure 3.** Fault damage zone from outcrop and drilling sampling. The fracture density decreases away from the fault core and down into the background level. (**a**) is modified from [7]. (**b**,**c**) are modified from [4]. (**d**) Based on the three-dimensional seismic variance properties, the FDZ widths for various beds are obviously different. Modified from [34]. (**e**) Log-log plots of FDZ width against displacement from previous studies. Modified from [6,35–40] and references therein.

## 3. Materials and Methods

### 3.1. Principle of Fault Likelihood

Characterizing fractures by extracting seismic attributes has become a common technique for seismic interpretation [41–43], with the common methods for extraction including coherence (Cx; x denotes the generation number), fault likelihood (FL), ant tracking, automatic fault extraction (AFE), and Canny edge detection [44–46]. Among these methods, fault likelihood presents significant advantages in characterizing the contours of fault zones. Specifically, this method identifies and characterizes faults from the 3D seismic data volume based on the principle of maximum likelihood and can reflect the widths of fractures to a certain extent. It has been widely used in recent years [47,48]. Maximum likelihood is designed to determine a certain parameter value that maximizes the occurrence probability of a sample using the results of multiple experiments. The fault likelihood method scans the 3D seismic data along a certain strike and fault dip direction and then calculates the minimum similarity of the 3D seismic data of various sample points. Since this method fully considers the variations in the strike and dip direction of faults, it yields high sensitivity and accuracy in characterizing faults. This method has become a conventional method used to characterize faults based on seismic attributes and can qualitatively and semi-quantitatively characterize the fault width [49]. In addition, fault likelihood is good at detecting faults with small throws and sizes [50]. Hale [45] proposed a new method for processing 3D seismic data that provides images of fault likelihoods and corresponding fault strikes and dips. Subsequently, the method has been continuously improved and optimized and is widely used in petroleum exploration [51,52].

Based on the summary of the properties of the faults, this study described the faults in detail using the fault likelihood with the interpretation software Paradigm17. The specific process was as follows: The faults previously identified were classified according to their seismic characteristics. Each type of fault was treated as a voxel with information such as amplitude, dip angle, and strike. During the calculations, the sampling points of each type of fault were compared with all the sampling points in the seismic data volume to determine the similarity between them, among which the minimum similarity was taken as the analytical result of the fault likelihood. Then, global normalization was conducted for all the minimum similarities, and relevant parameter values were determined to allow voxels with close similarities to correlate with each other. As a result, the fault likelihood was determined, which can effectively identify the faults identified previously using geological methods.

### 3.2. Attribute Results

The Sichuan Basin has the most abundant shale gas in China. The study area was in the southern Sichuan Basin (Figure 4a). The Lower Silurian Longmaxi Formation shale in the southern Sichuan Basin deposits on deep-water shelf to shallow shelf. There is little variation in the lithology, mineral composition, and sedimentary structures of shales in the Luzhou block.

This study characterized the faults in the shales of the Lower Silurian Longmaxi Formation in the Fuji syncline of the Luzhou block using the fault likelihood. The Luzhou block is located in a broom-like low, steep fold belt in the southern Sichuan Basin, and the surface structures in this block primarily have a NE strike. The Upper Ordovician Wufeng Formation has complex bottom structures composed of NE-, NNE-, and nearly EW-trending structures (Figure 4b). The main body of this block consists of four high, steep, and narrow NE-trending faulted anticlinal zones and flat, broad synclinal zones sandwiched between the faulted anticlinal zones, forming a typical comb-like fold belt. Faults have developed on both flanks of the high and steep anticlines, forming ramp and backthrust structures. As shown in the attribute prediction map, the red and yellow zones denote areas with large-scale faults and medium- and small-scale faults, respectively. The map presents clear characteristics of the faults, which primarily exhibit NEE trending (Figure 4c).

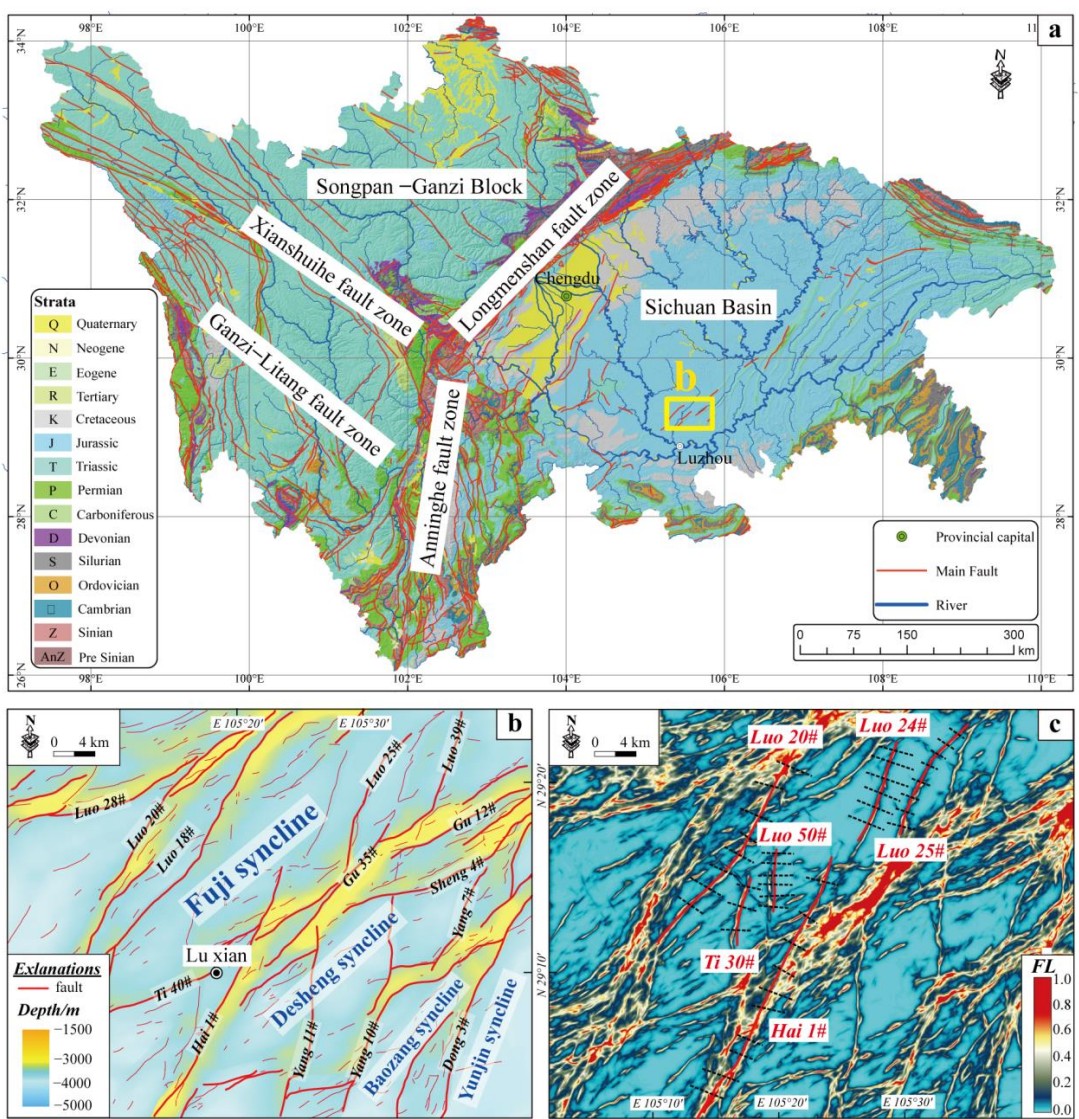

**Figure 4.** (**a**) Location of the study area (yellow rectangle) and generalized geological map of Sichuan Basin region. The strata data and fault data are from the digital Library of NGAC (http://www.ngac.org.cn/, accessed 20 July 2023). (**b**)Structural map of Upper Ordovician Wufeng Formation in Luzhou block in Sichuan Basin. (**c**) Fault likelihood map of Luzhou block. FL—fault likelihood value.

## 4. Discussion

### 4.1. FDZ Widths of Faults in the Fuji Syncline of the Luzhou Block

Faults are well developed in the shales of the Lower Silurian Longmaxi Formation in the southern Sichuan Basin. The influencing scopes of the FDZs of the concealed faults in the formation have long puzzled geologists and the exploration community. This study quantitatively characterized six major NEE-trending faults in the Fuji syncline based on 41 seismic profiles. Taking the fault Luo 24# as an example, the specific process is stated as follows: For this fault, six transverse seismic profiles with different lengths were employed (Figure 5) to uniformly control the variation in the fault throw along the fault strike, followed by detailed seismic interpretations. The fault throw along the strike was calculated (Figure 5), and the fault likelihood curves were then plotted. All the curves exhibited a peak of 0.4–0.7 at 0 m (Figure 6a), indicating a high occurrence probability of a fault, which was verified by several seismic profiles (Figure 5). As the distance from the fault increased, the probability in the curves decreased rapidly and was smaller than 0.05 at a distance of 500 m and above on both sides of the fault. Secondary peaks appeared locally

(distance: −400 m and 1600 m), and they might be caused by local disturbance of some small secondary faults (Figure 6a).

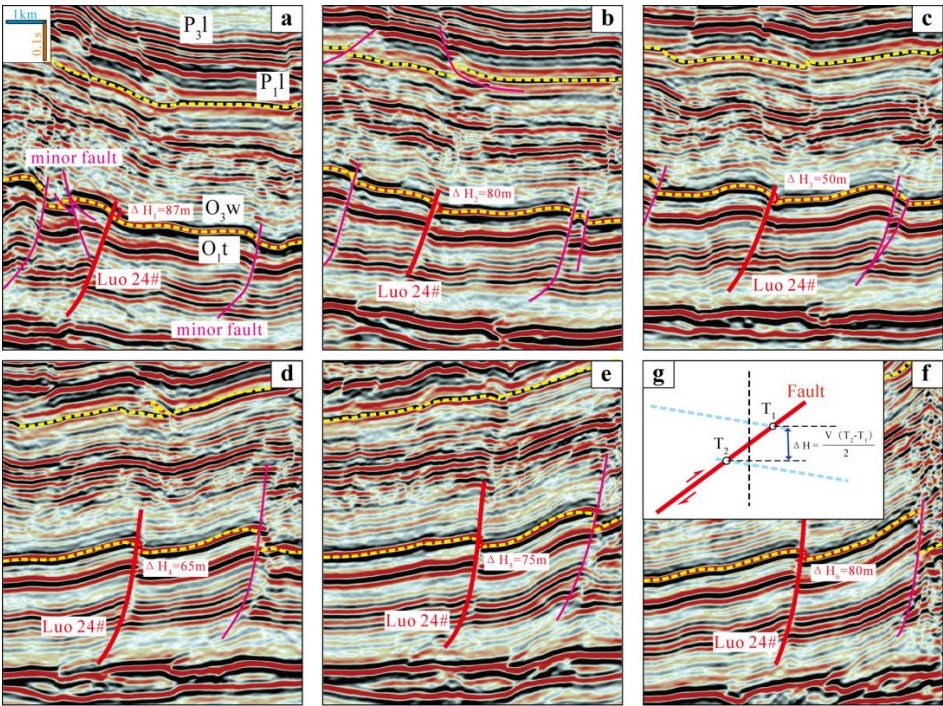

**Figure 5.** (**a**–**f**) The interpreted seismic profiles along Luo 24# fault in Fuji syncline in Luzhou block, Sichuan Basin (location see Figure 4). (**g**) Method for estimating fault throw along the seismic profile. The fault throw is calculated through difference in two-way time (TWT). $P_3l$—the Upper Permian Longtan Formation, $P_1l$—the Lower Permian Liangshan Formation, $O_3w$—the Upper Ordovician Wufeng Formation, $O_1T$—the Lower Ordovician Tongzi Formation. V is the average velocity of the layer cut by the fault. The yellow dotted line represents the stratigraphic boundary interpreted from the seismic profile.

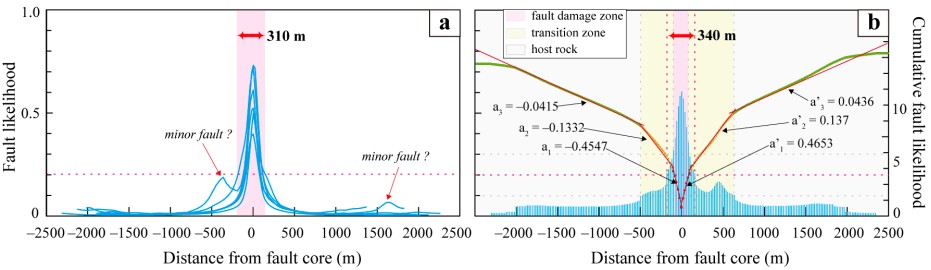

**Figure 6.** Relationship between fault likelihood value/(**a**) cumulative fault likelihood (**b**) and fault core distance of Luo 24# fault in Fuji syncline in Luzhou block, Sichuan Basin.

The fault likelihood is related to the degree of fracturing. In other words, a higher fault likelihood corresponds to a higher degree of fracturing and a higher fracture density. The selection of the fault likelihood for FDZs and surrounding rocks (i.e., the threshold) is crucial to the quantification of the FDZ width. The cumulative fault likelihood can be calculated using the cumulative fracture density. The position subjected to a shape decrease in the slope of the cumulative fault likelihood curve is considered the boundary between a FDZ and the surrounding rocks, and the fault likelihood at this position is considered the threshold. As shown by the variation in the slope of the cumulative fault likelihood curve (Figure 6b), the intersections of slopes with different gradients were present on each side of the Luo 24# fault. These intersections divided the area to be quantified into three zones, namely the intense fracture zone with medium-high fault likelihood (width: about

180 m), the transition zone with medium-low fault likelihood (width: about 860 m), and the surrounding rock zone with extremely low fault likelihood (Figure 6b). Theoretically, the threshold should be between 0.1 (corresponding to the boundary between the transition zone and the surrounding rocks) and 0.3 (corresponding to the boundary between the transition zone and the intense fracture zone). Such a threshold makes it possible to reduce the variation in the FDZ width caused by differences in the artificially defined boundaries of FDZs. Given that numerous background fractures occurred around the major fault zones in the study area, it was inappropriate to select an excessively low threshold for quantification. Based on extensive investigations and a comprehensive contrastive analysis of exposed fault damages, fractures in cores, and variances of seismic data, a fault likelihood of 0.2 as the threshold for the FDZ boundaries was selected in this study. Accordingly, the FDZ width was determined at 310–340 m (Figure 6).

The FDZ widths of five other faults (Luo 20#, Ti 30#, Luo 50#, Hai 1#, and Luo 25#) in the Fuji syncline were also determined in the same way. The average FDZ widths of the five faults were determined based on a threshold of 0.2 (Figure 7). The Ti 30# (Figure 7b) and Luo 50# (Figure 7c) faults showed relatively simple fault likelihood curves, indicating that their FDZs were each controlled by a single major fault. In contrast, the Luo 20# (Figure 7a), Hai 1# (Figure 7d), and Luo 25# (Figure 7e) faults exhibited relatively complex fault likelihood curves. Their curves presented several secondary peaks, suggesting that their FDZs might be complex multi-core structures due to the superposition of secondary faults. In addition, Luo 25# cuts other faults, so the curves may contain the older fractures. The FDZ width was about 1220 m (Figure 7d). The FDZ width of the Luo 25# fault was analyzed as follows. For the northern section of this fault along several seismic profiles, the fault likelihood curve primarily exhibited a single peak, and its FDZ width was determined at about 250 m. As the fault was obliquely connected to the NE-trending structures southward, the fault likelihood curve became increasingly complex, presenting several secondary peaks. This result indicates that the superposition of secondary faults significantly increases the width of the fault likelihood curve (Figure 7e). Consequently, the FDZ width exceeded 450 m. Overall, the FDZs widths of the six major faults in the Fuji syncline were between 240 and 1220 m.

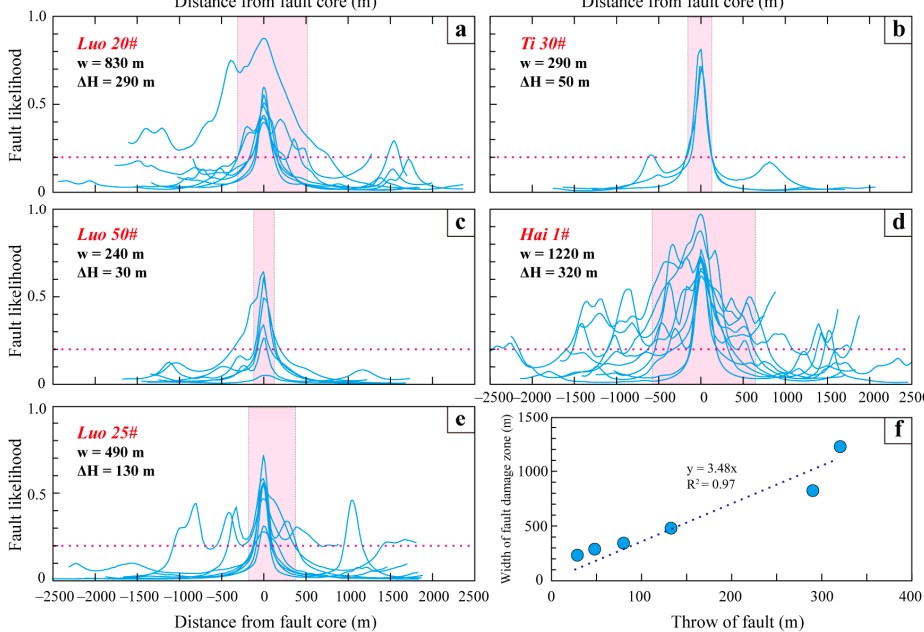

**Figure 7.** (**a–e**) Fault likelihood curve and width of fault damage zone in Fuji syncline in Luzhou block and (**f**) width of damage zone plotted against fault throw.

In addition, the FDZ width was affected by the fault throw, which varies along the fault strike. According to the seismic profile survey (Figure 8a–e), the Luo 20# fault had

a throw of 396 m, 271 m, 133 m, 122 m, and 182 m from north to south. Therefore, the fault throw was relatively high in the northern section and gradually decreased southward, reflecting that the Luo 20# fault grew from north to south. In contrast, the Hai 1# fault had a throw of 97 m, 83 m, 313 m, 414 m, 515 m, and 497 m from north to south. Therefore, the fault throw was less than 100 m in the northern section but increased significantly in the middle and southern sections (Figure 8f–k), reflecting the south-to-north growth of this fault. Many previous studies have shown that there is a significant positive correlation between the FDZ width and the fault scale and have plotted a relationship chart between the fault throw and the FDZ width (Figure 3e). Therefore, the reliability of the results of this study can be verified by projecting the data on the throw and FDZ width of faults in the Fuji syncline onto the relationship chart. As shown by the verification results, the FDZ widths quantified based on the curves of the fault likelihood varying with the distance from the fault are consistent with the intrinsic relationship between the FDZ width and the fault scale (Figure 3e).

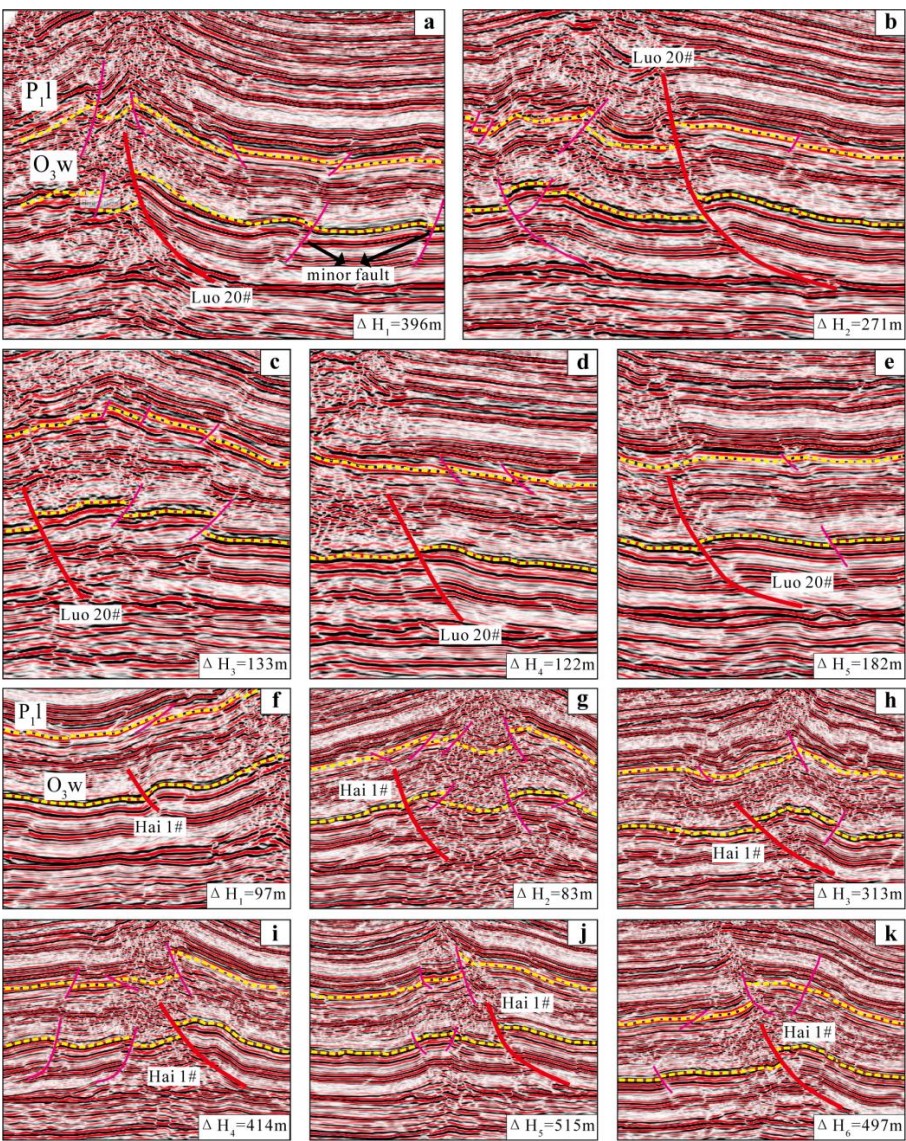

**Figure 8.** The interpreted seismic profiles along Luo 20# (**a–e**) and Hai 1# (**f–k**) in Fuji syncline in Luzhou block, Sichuan Basin (location see Figure 4). $P_1l$—the Lower Permian Liangshan Formation, $O_3w$—the Upper Ordovician Wufeng Formation. V is the average velocity of the layer cut by the fault. The yellow dotted line represents the stratigraphic boundary interpreted from the seismic profile.

*4.2. FDZ Widths Correlation with Fault Throws*

The throws of different-scale faults in the Fuji syncline and the average FDZ widths of these faults characterized using the fault likelihood were projected and linearly fitted. As shown by the results, the FDZ widths were about 3.5 times the fault throws, with a correlation coefficient of 0.97 (Figure 7f), indicating that the FDZ width is closely related to the fault throw. Generally, the fault throw can reflect the fault scale and determine the FDZ width. Since the fault throw can be easily determined in actual explorations, the FDZ width of a seismic-scale fault can be quickly and directly constrained using the quantitative relational expression between the fault throw and the FDZ width. However, the developmental degree of the fractures in faults on a sub-seismic scale and below and the FDZ width of these faults should be determined based on the improvement in the resolution of seismic data and the statistics of fractures in cores. Moreover, with an increase in data on fault samples, the ratio of the FDZ width to the fault throw may vary to some extent and thus should be adjusted according to actual applications.

**5. Conclusions**

1. FDZs are stress–strain concentration zones formed due to the action of faulting on surrounding rocks, and the influencing scope of faulting can be effectively reflected by the developmental degree of fault damages. In this study, quantification of FDZ width using the threshold of the fault likelihood was attempted. Based on the points subjected to the changes in the slope of the cumulative fault likelihood, as well as the seismic profiles for structural interpretations, the fault likelihood of 0.2 was selected as the threshold for FDZ width. Consequently, the faults in the Fuji syncline were determined to have a FDZ width of 240–1220 m.

2. For the faults in the shales of the Longmaxi Formation in the Luzhou block of the southern Sichuan Basin, the FDZ widths were 3.5 times the fault throws. This relational expression can be conveniently used in actual applications and serve as a guide for the exploration and exploitation of shale gas in the basin. However, it is necessary to further consider the effects of the developmental degree of fractures of the faults on a sub-seismic scale and below on shale gas preservation conditions and well placement.

**Author Contributions:** Conceptualization, L.Z. and J.L.; methodology, J.L. and Z.L.; software, J.L.; validation, L.Z.; formal analysis, L.Z.; investigation, S.W. and K.T.; resources, L.Z.; data curation, J.L.; writing, L.Z.; draft preparation, L.Z.; writing—review and editing, L.Z.; supervision and project administration, J.L. All authors have read and agreed to the published version of the manuscript.

**Funding:** This research was funded by the National Natural Science Foundation of China (Grant no. 42001012), the CAS 'Light of West China' Program (Grant no. Y9R2140145), and the Youth Fund of the Institute of Mountain Hazards and Environment, CAS (Grant no. Y9K2100100).

**Institutional Review Board Statement:** Not applicable.

**Informed Consent Statement:** Not applicable.

**Data Availability Statement:** Not applicable.

**Conflicts of Interest:** The authors declare no conflict of interest.

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
