# Peer review of "Quantifying the Widths of Fault Damage Zones Based on the Fault Likelihood: A Case Study of Faults in the Fuji Syncline of the Luzhou Block, Sichuan Basin, China"

_sustainability, doi:10.3390/su151511771_

Round 1

Reviewer 1 Report

The study of the geological structure characteristics of the complex structural zones around the Sichuan Basin, such as faults, folds, shear zones, is crucial to the study of Shale gas preservation conditions. This manuscript takes Luzhou Block in Sichuan Basin as an example to quantify the widths of fault damage zones based on the fault likelihood. The authors have done some interesting work. These conclusions provide a better working procedure for quantifying the constraints of geological structures on Shale gas preservation conditions. I propose to revise and publish. 

A few practical suggestions are as follows: 

1.The pattern of the fault core area in Figure 1 (such as solid angular gravel particles) needs to be annotated and explained in the figure;

2.In Figure 1, there are two arrows indicating faults. Do the authors mean that this image indicates the development of two important faults?

3. How to distinguish fault core and fault damage zone? And whether the fault core is removed when counting the width of the fault damage zones?

4.Does the main body of the manuscript require the use of Fig or Figure? Please unify the usage methods of the two by the author. I have annotated it in the attachment.

5. Line 131 “This is a figure. Schemes follow the same formatting.” should be removed.

6. Headings 3.1 (line 164) and 3.2 (line 193) are duplicated, please confirm.

7.What exactly does the abbreviation FL in Figure 4b refer to? 

8. How can the fault throw along the seismic profile Figures 5 and 8 be estimated? Can you provide a calculation method?

9.Please refer to the attachment I uploaded for other modification suggestions.

 Minor editing of English language required

Reviewer 2 Report

Dear author, I am glad to receive your manuscript. In response to your article Quantifying the widths of fault damage zones based on the fault likelihood: A case study of faults in the Fuji syncline of the Luzhou block, Sichuan Basin, China, the following suggestions have been made

1The author used the fault likelihood method to estimate the width of 18 fault failure zones (FDZs), which should be verified through experiments or numerical simulations.

2The analysis of "Basic structures of FDZs " and other aspects lacks theoretical basis, and mathematical models should be added to enhance the reliability of the research.

3Insufficient analysis of the "Principle of the fault like hood" in domestic and international research status.

4The specific basis for FDZ widths of 21 about 240‒1220 mis not reflected in the text.

5The images in the text should have corresponding explanations.

Some of sections need to be improved.

Reviewer 3 Report

Dear authors, 

I have got to review a manuscript entitled “Quantifying the widths of fault damage zones based on the fault likelihood: A case study of faults in the Fuji syncline of the Luzhou block, Sichuan Basin, China” by Lu Zeng, Jinxi Li,, Shihu Wu, Kailin Tong and Zhiwu Li.

The authors took up the issue of fault damage zone (FDZ), but their approach is conditioned by the exploration and exploitation of shale gas. As they mentioned in the abstract the study “estimated the widths of 18 fault damage zones (FDZs) using the fault likelihood.”

The actual object of the research were the faults in the Fuji syncline of the Luzhou block and they authors applied the approach described in the papers they cited, but in different areas. The content may be interesting (motivating?) for many engineers involved in shale gas extraction.

Some remarks:
Abstract.

“Faults are critical to the preservation or destruction of shale gas” You mean its concentration? Deposit?

Anyway such introduction of the abstract is OK but in the next you mention about the area of the study and as the next a method is explained. There is a gap between. Just add information about a real object of your study: the faults, their number and significance in the Silurian formation you mentined in the abstract. The formation is not the object of your study, the faults are! There are results of the study as the next, quite detailed ones. Between them:

“The degree of influence of the FDZs is negatively correlated with their distance from the faults. In other words, a greater distance from a fault is associated with a weaker influence and a smaller fault likelihood”.

did anyone expect a different pattern?

 Between us: it is not a good idea to make an abstract where 50% is devoted to the results but the scientific context, the scientific relevance of the research is missed.

The mentioned results should be formulated more general.

 Introduction

It looks OK, there are quite many references to earlier works in the text. There is an explanation of area study “The southern Sichuan Basin is one of the areas with the most abundant shale gas 63 resources in China since the Wufeng-Longmaxi…” but I would as for the map showing the location of this area. China is very, very big country so it’d be nice to have an idea of the area just "at the first look". In fact two maps would be OK: 1. location of the area with the boundary of the country as the background. 2. The second required map is a classical map: topographical or any map showing the terrain. In this logical sequence, another map, the geological one, is already shown (further on, never mind). It is clear, legible and complements the knowledge about the discussed space. Although the authors refer to works on geology, it would be appropriate to write a short geological (“consists primarily of fault gouge, cataclasites, and 81 breccias, with pseudotachylites possibly occurring in the fault core locally” – is not enough), and especially geomechanical, characterization of faulted rocks. In my opinion for as a scientific article, the text of the work has been strictly limited to basic information, which is more like a technical note. And the introduction chapter is an example of that.

The next chapter titled “Basic structures of FDZs” gives more comprehensive facts which are important for the further analysis. I am impressed with Figure 1. The figure is very informative and demonstrative. I wish that the author of this masterpiece would use his skills in the graphic presentation of the research results (3D distribution). There are also many valuable data and facts about the discussed problem. But finishing this chapter the author should consider how to make a link to the next text in the next chapter (“the authors examined the relations … in the condition of…. and it is presented further). It looks like a cut end story to me.

The next chapters (Materials and Methods and Discussion) are OK. I did not find any mistakes or errors in the analysis. The analytical procedures were used substantively to demonstrate relationship between the parameters: fault likelihood value (a)/cumulative fault likelihood (b) and fault core distance in Fuji syncline in Luzhou block, Sichuan basin.

I have a problem with the figure 8 in the conclusions. Is this a place to demonstrate anything in the form of figures? Remove Fig. 8 from the conclusions chapter.

Or rather it is a platform to announce research success with the conclusions described, possible consequences for existing knowledge or actions taken in engineering practice, reflection on future research (Sustainability has several unique features: Manuscripts regarding research proposals and research ideas are welcome).

At the first look there is something wrong: in the abstract we have 3 results and there are only two in this chapter. Anyway it must be changed and expanded: 1. write was the idea of your research and what kind of need was at the source of addressing this problem, 2. in short words ("military style"), please give the results in points (it's nice that the authors give values, it should always be like this) – as you have but with shorter explanation. 3 Try to find and to express a significance of your results and how the next research in future should be carried out. That will be a chapter called “The conclusions”.

Some more general remarks and suggestions

As I see the authors concentrated on the seismic profiles and a statistical approach, so they a bit missed a bit the pattern of this study. First of all, I would suggest a geological methodology: the analytical part should be preceded by figures which would explain much better the problem. So a topographical map of the study area and geological profiles much better would demonstrate than large descriptions. The minimum to be done (added) – the maps: 1. location of the area with the boundary of the country as the background. 2. a classical map: topographical or any map showing the terrain

I have several essential questions/remarks which deal with the fundamental conception of your research

1. “Statistical analysis of 2D areal density: ….These methods have their respective applicability …  and should be employed according to the actual situation..”. have you considered a 3D analysis or just spatial approach to the distribution of the analyzed parameters?

2. Have you considered a geostatistical analysis for relationships between mechanical and petrophysical properties of deformed rocks? My suggestion is to create a local coordinate system along strike line of a particular fault to find correlations.

3. Have you any idea how to expand the research and to involve a geostatistical approach with different parameters?

In my opinion, the manuscript is in this form (data, facts expressed in very briefly) more a kind of technical note than a scientific work, but taking into account the discussed content it is still a scientific work. So, please expand it according to suggestions I expressed in my review.

In spite of this it represents a good quality and very good level of the language which is used by the authors, a good style, it is very understandable. Undoubtedly, all figures are legible and clear and do not require any corrections.

There are only a few corrections in the text to be made by the author, so please improve the manuscript and look at the attached pdf file with handwritten corrections that will improve the quality of the text.

 Because this research work is interesting and important, I decided to have control over the quality of the publication, so I have clicked "Accept after minor revision" option...

with kind regards
a reviewer
